# A partial oviraptorosaur skeleton suggests low caenagnathid diversity in the Late Cretaceous Nemegt Formation of Mongolia

**Gregory F. Funston**[1]*, **Philip J. Currie**[2], **Chinzorig Tsogtbaatar**[3],
**Tsogtbaatar Khishigjav**[4]

**1** School of GeoSciences, University of Edinburgh, Edinburgh, United Kingdom, **2** Department of Biological Sciences, University of Alberta, Edmonton, AB, Canada, **3** NC Museum of Natural Sciences, Department of Biological Sciences, NC State University, Raleigh, NC, United States of America, **4** Institute of Paleontology, Mongolian Academy of Sciences, Ulaanbaatar, Mongolia

* Gregory.Funston@ed.ac.uk

**Data Availability Statement:** All data used in the study are available in the paper, except for precise coordinates of the fossil localities, which are withheld to protect the sites. Precise coordinates

## Abstract

The Nemegt Formation of the Gobi Desert of Mongolia has produced one of the most abundant and diverse oviraptorosaur records globally. However, the caenagnathid component of this fauna remains poorly known. Two caenagnathid taxa are currently recognized from the Nemegt Formation: *Elmisaurus rarus* and *Nomingia gobiensis*. Because these taxa are known from mostly non-overlapping material, there are concerns that they could represent the same animal. A partial, weathered caenagnathid skeleton discovered adjacent to the holotype quarry of *Nomingia gobiensis* is referable to *Elmisaurus rarus*, revealing more of the morphology of the cranium, mandible, pectoral girdle, and pubis. Despite metatarsals clearly exhibiting autapomorphies of *Elmisaurus rarus*, overlapping elements are identical to those of *Nomingia gobiensis*, and add to a growing body of evidence that these taxa represent a single morphotype. In the absence of any positive evidence for two caenagnathid taxa in the Nemegt Formation, *Nomingia gobiensis* is best regarded as a junior synonym of *Elmisaurus rarus*. Low caenagnathid diversity in the Nemegt Formation may reflect broader coexistence patterns with other oviraptorosaur families, particularly oviraptorids. In contrast to North America, competition with the exceptionally diverse oviraptorids may have restricted caenagnathids to marginal roles in Late Cretaceous Asian ecosystems.

## Introduction

The Nemegt and Ingenii Höövör Basins in the Gobi Desert of southwestern Mongolia are home to some of the richest Upper Cretaceous fossil sites in the world. Here, the Djadokhta, Baruungoyot and Nemegt Formations are exposed in a series of grabens and half grabens, forming discrete patches of extensive outcrops [1–5]. These exposures have produced one of the most diverse and best preserved faunal records of any Late Cretaceous ecosystem [6, 7]. The theropod component of this fauna is particularly well known, in contrast to other parts of the world, where the delicate bones of theropods are rarely preserved [8, 9]. From the Nemegt

are accessioned with the specimens at the Institute of Paleontology, Mongolian Academy of Sciences in Ulaanbaatar, Mongolia, and are available upon request by email (ipt@mas.ac.mn) or phone [+(976) 70118283].

**Funding:** Funding to GFF for travel and research in Mongolia was provided by a Michael Smith Foreign Study Supplement from NSERC (https://www.nserc-crsng.gc.ca/index_eng.asp) and Vanier Canada (https://vanier.gc.ca). GFF is funded by the Royal Society [Grant NIF\R1\191527] (https://royalsociety.org/). PJC is funded by the Natural Sciences and Engineering Research Council of Canada [Grant RGPIN-2017-04715] (https://www.nserc-crsng.gc.ca/index_eng.asp). The funders had no role in study design, data collection and analysis, decision to publish, or preparation of the manuscript.

**Competing interests:** The authors have declared that no competing interests exist.

Formation alone, a diverse assemblage of alvarezsaurids [10], dromaeosaurs [11], ornithomimids [12, 13], oviraptorosaurs [14, 15], therizinosaurs [16], troodontids [17], and tyrannosaurs [18, 19] is known, comprising nearly 250 associated skeletons [7, 20]. Whereas tyrannosaurs and ornithomimids form the bulk of the collected specimens, the oviraptorosaurs are the most diverse component of the fauna, known from at least nine genera [14, 15]. Unlike elsewhere [21], these genera represent each of the three Late Cretaceous families of oviraptorosaurs: avimimids, caenagnathids, and oviraptorids.

Despite the richness of the oviraptorosaur record in the Nemegt Formation, the caenagnathids of the Nemegt remain relatively poorly known. Osmólska [22] described three tarsometatarsi, one of which was associated with other parts of the skeleton, and she erected the new taxon *Elmisaurus rarus* Osmólska 1981 [22] for this material, reflecting its rarity. Barsbold et al. [23] described a relatively complete postcranial skeleton—including the first non-avian example of a pygostyle—collected in 1994 at Bügiin Tsav by the Japan-Mongolia Joint Paleontological Expedition. Subsequently, they described the skeleton in more detail, establishing a new taxon (*Nomingia gobiensis* Barsbold et al. 2000 [24]), and noting several morphological similarities to caenagnathids from North America [24]. Unfortunately, the lack of overlapping material made comparison with *Elmisaurus rarus* impossible. A caenagnathid identity for *Nomingia gobiensis* was also supported by Maryańska et al. [25], Osmólska et al. [26], and Sullivan et al. [27]. However, in the intervening years, most phylogenetic analyses have allied *Nomingia gobiensis* with oviraptorids, rather than caenagnathids [28–32]. Funston et al. [14] discussed this issue, noting that specimens not currently incorporated into phylogenies showed that each of the characters uniting *Nomingia gobiensis* with oviraptorids were also present in caenagnathids. Indeed, an updated phylogeny including these specimens recovers *Nomingia gobiensis* as a deeply-nested caenagnathid [15, 33].

Since the initial descriptions of *Elmisaurus rarus* and *Nomingia gobiensis*, few new caenagnathid specimens have been discovered from the Nemegt Formation. Persons et al. [34] referred an isolated pygostyle to *Nomingia gobiensis*, but they considered this taxon simply an 'advanced oviraptorosaur' rather than supporting either a caenagnathid or oviraptorid interpretation. In 2016, two important studies described new material. Currie et al. [35] described additional specimens of *Elmisaurus rarus*, showing that it had an unusual frontal that probably accommodated a crest, and revealing more of the fore- and hindlimbs. Tsuihiji et al. [36] described the first caenagnathid dentaries from the Nemegt Formation, allowing for critical comparisons to North American caenagnathids, which are overwhelmingly represented by the robust, fused dentaries [37, 38]. However, lack of association to other parts of the skeleton meant that it was ambiguous whether these dentaries pertained to *Elmisaurus rarus* or possibly *Nomingia gobiensis*. Funston et al. [14] reviewed the oviraptorosaurs of the Nemegt Basin, and commented on the possible synonymy of *Elmisaurus rarus* and *Nomingia gobiensis*. Indeed, many of the new elements (vertebrae, pubes, tibiae) described by Currie et al. [35] are identical to those of *Nomingia gobiensis*, despite clearly being referable to *Elmisaurus rarus* on the basis of the associated tarsometatarsi.

In 2018, the holotype quarry of *Nomingia gobiensis* at Bügiin Tsav was revisited by a joint Italian-Canadian-Mongolian expedition. Although no new bones were found from the type specimen, a second fragmentary caenagnathid skeleton was found as float weathering from the same horizon approximately 12 metres south of the quarry (Fig 1). Portions of the tarsometatarsus allow this individual to be identified as an immature *Elmisaurus rarus*, but similarities elsewhere in the skeleton add to a growing body of evidence that *Nomingia gobiensis* is the junior synonym of *Elmisaurus rarus*.

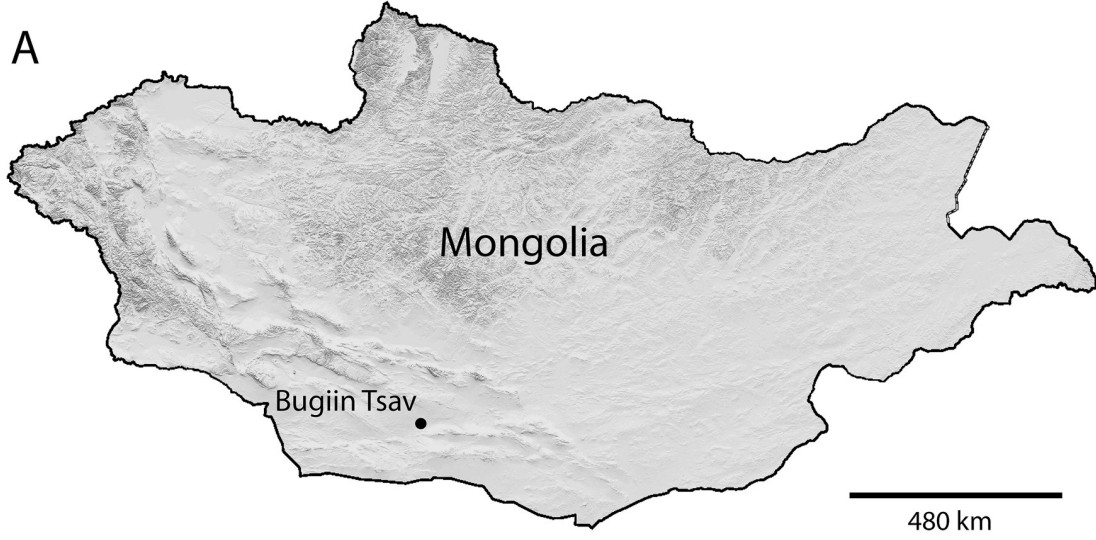

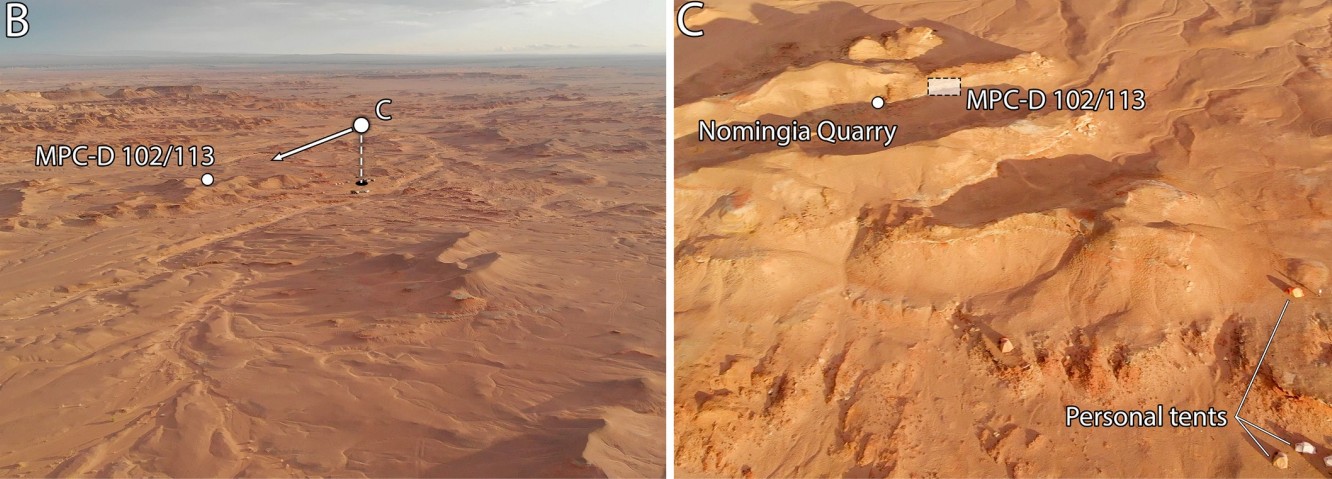

**Fig 1. Location of the Bügiin Tsav locality and "*Nomingia gobiensis*" type quarry. A**, Topographic of Mongolia showing the Bügiin Tsav region in the Ingenii Höövör Basin. Map data from USGS National Map Viewer (Public Domain). **B**, Aerial image of Bügiin Tsav looking southeast, showing the site where MPC-D 102/113 was collected, and the orientation of the aerial image (C). **C**, aerial image of the site of MPC-D 102/113 from above the 2018 Camp site, showing the proximity of the site to the type quarry of "*Nomingia gobiensis*". Note single-person tents in bottom right for scale.

## Materials and methods

Palaeontological fieldwork was conducted under the necessary permits, issued by the Institute of Paleontology of the Mongolian Academy of Sciences, Ulaanbaatar, Mongolia. Precise GPS coordinates of the site are accessioned with the specimen at the Institute of Paleontology of the Mongolian Academy of Sciences, Ulaanbaatar, Mongolia, where they are available upon request. The specimen MPC-D 102/113 was collected as surface float and was repaired using cyanoacrylate. Aerial images of the site were taken from video frames acquired using a DJI Mavic Air with a gimbal-stabilized 24 mm-equivalent 12 MP camera. In 2018 when the flights were conducted, no additional permits or permissions were required, although regulations were introduced 12 June 2019 that require permission to be obtained before flying. Measurements were taken with digital calipers to an accuracy of ±0.5 mm. The specimen was photographed using a Nikon D7200 digital camera equipped with a Nikkor 18–140 mm lens and a

lens-mounted ring light. Photos were processed in Adobe Photoshop CC to remove the background, and where adjustments were made to contrast, brightness, or colour balance, these were applied to the whole image. Mongolian locality names follow Benton [39].

### Institutional abbreviations

**MPC**, Institute of Paleontology, Mongolian Academy of Sciences, Ulaanbaatar, Mongolia; **TMP**, Royal Tyrrell Museum of Palaeontology, Drumheller, Alberta, Canada; **ZPAL**, Institute of Paleobiology, Polish Academy of Sciences, Warsaw, Poland.

## Results

### Systematic palaeontology

Dinosauria Owen 1842 [40]

Theropoda Marsh 1881 [41]

Oviraptorosauria Barsbold 1976 [42]

Caenagnathidae Sternberg 1940 [43]

*Elmisaurus rarus* Osmólska 1981 [22]

**Newly referred material.**   MPC-D 102/113, partial skeleton comprising possible partial postorbital, partial angular, partial cervical vertebra, nearly complete scapulocoracoid, nearly complete left pubis, partial right pubis, partial astragalus, and partial left tarsometatarsus.

**Horizon and locality.**   Nemegt Formation (?Maastrichtian), Bügiin Tsav, Gobi Desert, Mongolia. Collected approximately 12 m south of the type quarry of *Nomingia gobiensis* (MPC-D 100/119). Precise GPS coordinates for both sites are available to researchers upon request.

### Description

The material was collected as an assemblage of float fragments concentrated in a small area and derived from a single fossiliferous horizon. This deposit also produced isolated elements pertaining to a dromaeosaur, turtles, and other indeterminate small–medium sized vertebrate remains. The float material can be recognized as pertaining to the same individual not only by the consistent size and tight articulation of the elements, but also by the distinctive weathered bone surface, which contrasts with the pristine surface conditions of other elements recovered from the fossiliferous horizon. This weathering is most consistent with pre-burial weathering and corresponds to Stages 1–2 of Behrensmeyer [44], which indicates that the elements share a unique taphonomic history. Furthermore, none of the elements are inconsistent with those known for other caenagnathids or *Elmisaurus rarus*, which suggests they are not chimeric. The most reasonable interpretation is that these elements represent the remains of a single skeleton that was extensively weathered before being interred in a point bar deposit.

**Possible postorbital.**   A small triangular fragment of bone (Fig 2A–2E) cannot be identified with certainty, although it appears to have surrounded one of the cranial fenestrae. It is similar to the postorbitals of other oviraptorosaurs, although it could be a heavily worn fragment of another cranial bone. The bone tapers towards one side, presumably dorsal, and is broken on the opposite side. The presumable anterior edge of the bone is straight, whereas the opposite edge is rounded into a tab-like flange. On what is presumably the lateral surface of

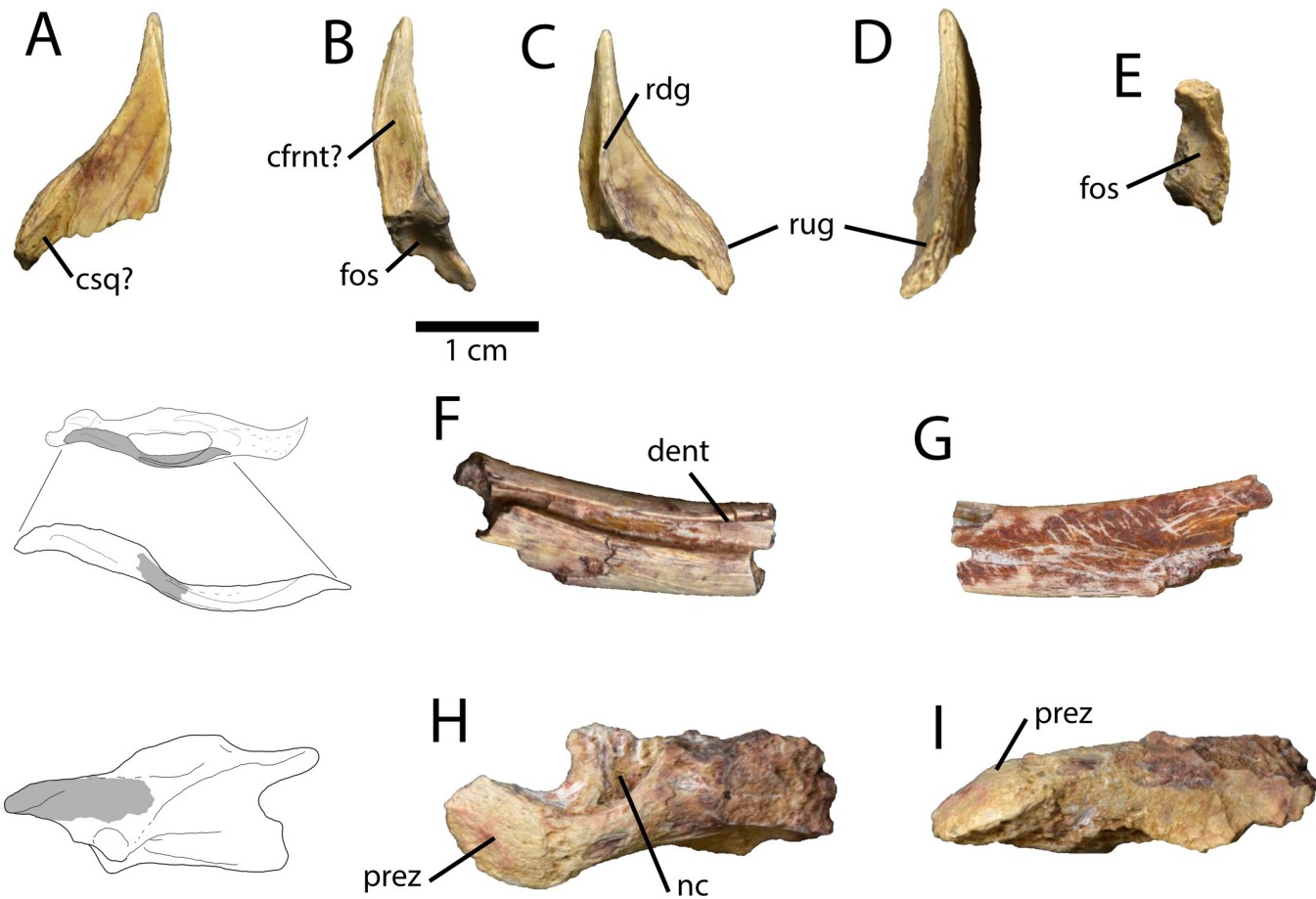

**Fig 2. Cranial and axial elements of *Elmisaurus rarus* (MPC-D 102/113). A–E,** Possible right postorbital in presumed lateral (A), anterior (B), medial (C), posterior (D), and ventral (E) views. **F**, Right angular in lateral view (F) and medial (G) views, with interpretive illustration of *Chirostenotes pergracilis* (TMP 2001.012.0012) showing the approximate location of the fragment. **H, I**, Partial cervical neural arch in dorsal (H) and left lateral (I) views, with interpretive illustration of an anterior cervical vertebra of *Epichirostenotes pergracilis* (ROM 43250) showing the approximate location of the fragment. **Abbreviations: cdent**, contact groove for dentary; **cfrnt?**, presumed contact surface for frontal; **csq?** presumed contact surface for squamosal; **fos**, fossa; **nc**, neural canal; **prez**, prezygapophysis; **rdg**, ridge; **rug**, rugosity.

this flange, there is a small groove into which another bone may have inserted (Fig 2A: csq?). The opposite, presumably medial surface of the bone has a longitudinal ridge that separates it into two concave portions (Fig 2C: rdg). The concavity closest to the straighter edge is smaller and shallows towards the tapering end of the bone. The other concavity is larger and has two shallow grooves extending perpendicular to the medial ridge. The end of the bone opposite the tapering process is broken, but there is a region of finished surface that indicates either the bone was hollow, or that it branched into multiple processes that are broken. Although the bone is complex, it cannot be positively identified because caenagnathid crania are so poorly known and there is little comparative material. The most likely option is that it represents an unusual postorbital, but it is possible that it represents a heavily worn fragment of the frontal, the lacrimal, the maxilla, or a palatal bone.

**Dentary and angular.** A small fragment of bone (Fig 2F–2G) represents part of the right mandible below the external mandibular fenestra. Most of this fragment is comprised of the angular, which has a deep, posteriorly tapering groove on its lateral surface (Fig 2F: cdent). Within this groove, a small portion of the splint-like posteroventral ramus of the dentary is

preserved. This groove and the portion of the dentary occupying it are dorsoventrally narrower than the corresponding features in *Chirostenotes pergracilis* Gilmore 1924 [45–47], where the groove on the angular shallows and broadens posteriorly. Together, these two bones bow ventrally, as is the case in *Anzu wyliei* Lamanna et al. 2014 [28], *Apatoraptor pennatus* Funston and Currie 2016 [29], and *Chirostenotes pergracilis* [45]. However, the angular of *Caenagnathus collinsi* Sternberg 1940 [43] appears to be less ventrally curved [37, 43]. The medial surface of the angular of MPC-D 102/113 has a slight depression, resulting in an hourglass-like cross-section of the bone.

**Cervical vertebra.**   A small portion of a cervical vertebra (Fig 2H–2I) was recovered. It includes the base of the neural arch and the left prezygapophysis. The prezygapophysis faces anterodorsally and is rounded in dorsal view (Fig 2H). Its medial edge extends posteriorly as a ridge and curves medially to overhang a pocket above the neural canal. It is difficult to determine which part of the neck the prezygapophysis is derived from, but based on the great depth of the fossa overlying the neural canal and the large proportion of the prezygapophysis occupied by the articular area, it likely represents an anterior vertebra.

**Scapulocoracoid.**   The left scapula and coracoid (Fig 3A–3C) are relatively complete, but the scapula is missing its distal end. The bones are unfused, contrasting the condition in *Anzu wyliei* [28] but consistent with *Apatoraptor pennatus* [29], *Elmisaurus rarus* [35], and apparently *Chirostenotes pergracilis* [48], based on the isolated coracoid of TMP 1979.020.0001. However, this may be the result of ontogenetic differences between the known specimens. The scapula has a transversely thick blade that curves medially and ventrally, contrasting the straighter and more gracile scapular blades of most oviraptorids [15, 26]. The acromion is damaged but appears to have been narrow and directed anterolaterally. The long anterior

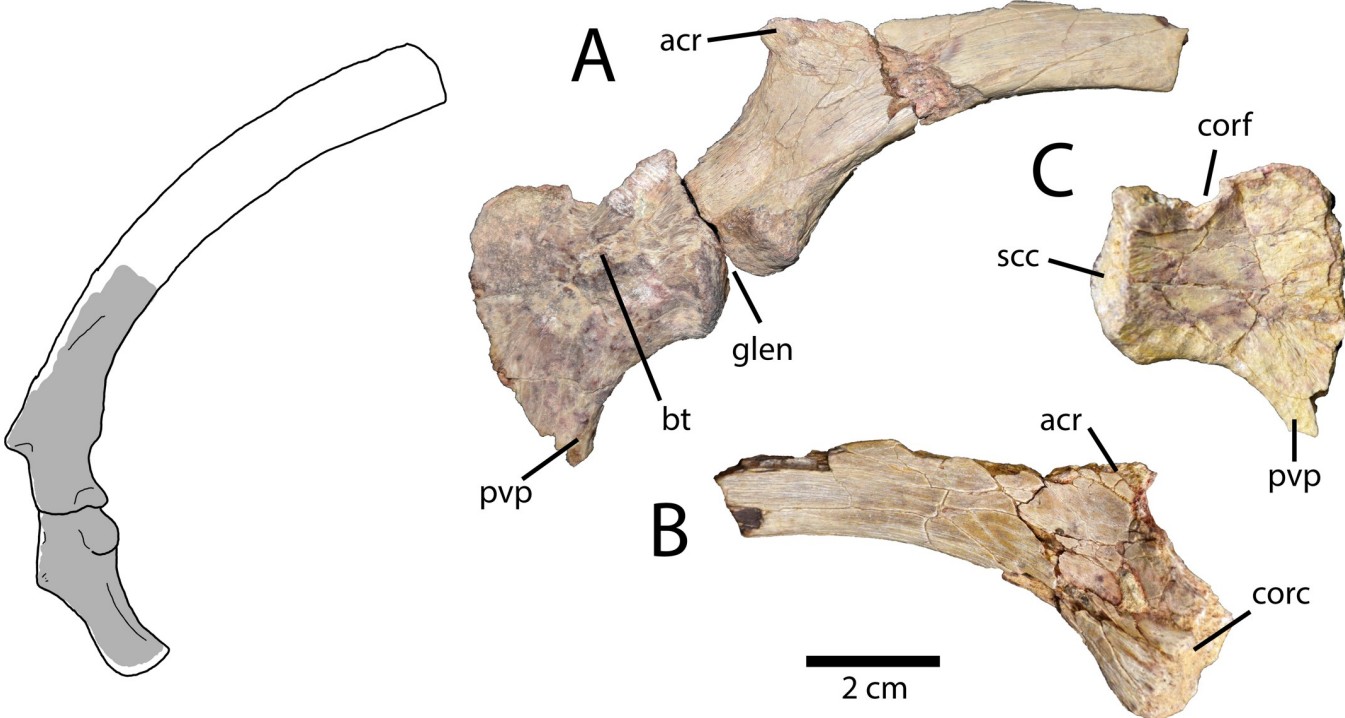

**Fig 3. Pectoral girdle of *Elmisaurus rarus* (MPC-D 102/113). A,** Left scapulocoracoid in lateral view. **B,** Left scapula in medial view. **C,** Left coracoid in medial view. Interpretive illustration of *Apatoraptor pennatus* (TMP 1993.051.0001) shows the approximate locations of the preserved portions. **Abbreviations: acr**, acromion process; **bt**, biceps tubercle; **corc**, coracoid contact; **corf**, coracoid foramen; **glen**, glenoid; **pvp**, posteroventral process; **scc**, scapular contact.

portion of the scapula is downturned relative to the shaft, so the glenoid is situated more ante-riorly and ventrally than in other oviraptorosaurs. In lateral view (Fig 3A), the posterior edge of the glenoid is anterior to the level of the acromion process. In contrast, in oviraptorids (e.g. *Heyuannia yanshini* (Barsbold 1981) [49], *Oksoko avarsan* Funston et al. 2020 [15], *Rinchenia mongoliensis* (Barsbold 1986) [50], and the Zamyn Khond oviraptorid) the anterior edge of the glenoid is directly ventral to the apex of the acromion process. The scapular portion of the gle-noid of MPC-D 102/113 is small and circular, rather than elongate. It is slightly inclined to face laterally, although not as much as in *Apatoraptor pennatus* [29]. Just posterior to the gle-noid is a small nutrient foramen, and posterior to this, the scapula is rugose along its ventral edge. The scapula has a long contact with the coracoid, which is inclined about 45˚ anterodor-sal–posteroventral in lateral view, depending on the orientation of the scapula within the body. This results in a large area of bone that projects anteriorly between the glenoid and the acro-mion process (Fig 3A). The medial surface of this area (Fig 3B) is concave and is penetrated by several small foramina. Unlike the scapula of some oviraptorids (e.g. *Oksoko avarsan*; MPC-D 100/33), this area lacks a longitudinal groove conducting the vasculature from the coracoid foramen towards the blade of the scapula. However, this may be explained by the dorsal posi-tion of the coracoid foramen, which is situated above the dorsal edge of the scapula.

The coracoid (Fig 3A and 3C) is well preserved and is complete except for its dorsal and anteroventral edges. The glenoid is only slightly inclined relative to the posterior edge of the coracoid, which results in a more posterior orientation of the entire glenoid, rather than ven-tral. Just anterior to the coracoid portion of the glenoid there is a small depression, and the sur-face anterior to this is rugose. The biceps tubercle is positioned relatively far dorsally for an oviraptorosaur, entirely dorsal to the glenoid (Fig 3A). It is large and mounded, but does not protrude as much as those of most oviraptorids [26]. The coracoid foramen is directly dorsal to the biceps tubercle (Fig 3C), rather than posterodorsal, and as noted previously, this situates it above the dorsal edge of the scapula. The posteroventral process curves posteriorly, although it is straighter than that of most oviraptorosaurs, as is also the case in *Apatoraptor pennatus* [29]. Its apex is broken. On the medial side of the coracoid, there are no fossae underlying the biceps tubercle, unlike the condition in *Oksoko avarsan* (MPC-D 100/33; Funston et al. 2020a).

**Pubis.**   Parts of both pubes are preserved (Fig 4A–4C), but the left is far more complete. The proximal end is nearly completely preserved, as is most of the shaft, but the pubic boot is missing. The iliac process of the head is eroded, exposing the medullary cavity, which suggests it had decomposed before burial. The acetabular portion is roughly trapezoidal, with the shorter side facing laterally, and is dorsally concave. The ischiadic contact is elliptical and its lateral edge protrudes slightly from the head of the pubis. There is a ventral groove underlying the ischiadic contact, but otherwise it does not protrude posteriorly (Fig 4A), which is also the case in *Anzu wyliei*, *Elmisaurus rarus*, and *Nomingia gobiensis* [14]. The medial surface of the pubic head is concave, and the ischiadic contact protrudes medially to form a posterior lip around this concavity (Fig 4B and 4C: pf), suggested to be a synapomorphy for caenagnathids [27]. However, this lip protrudes further medially than those of most other caenagnathids. The lateral surface of the proximal pubic head is rugose, whereas the bone further distally is more fibrous. The shaft of the pubis is also rugose, but posteriorly it is pitted and pock-marked, which suggests that this texture is the result of weathering before burial. The shaft of the pubis is relatively straight throughout its length in lateral view, although it is anteriorly concave as in all oviraptorosaurs. The entire proximodistal length of the pubic apron is preserved, but its medial edge is worn, which reduces its transverse width. Regardless, rearticulation with the right pubis indicates that the pubic apron would have been transversely narrow (Fig 4C), as is the case in all oviraptorosaurs.

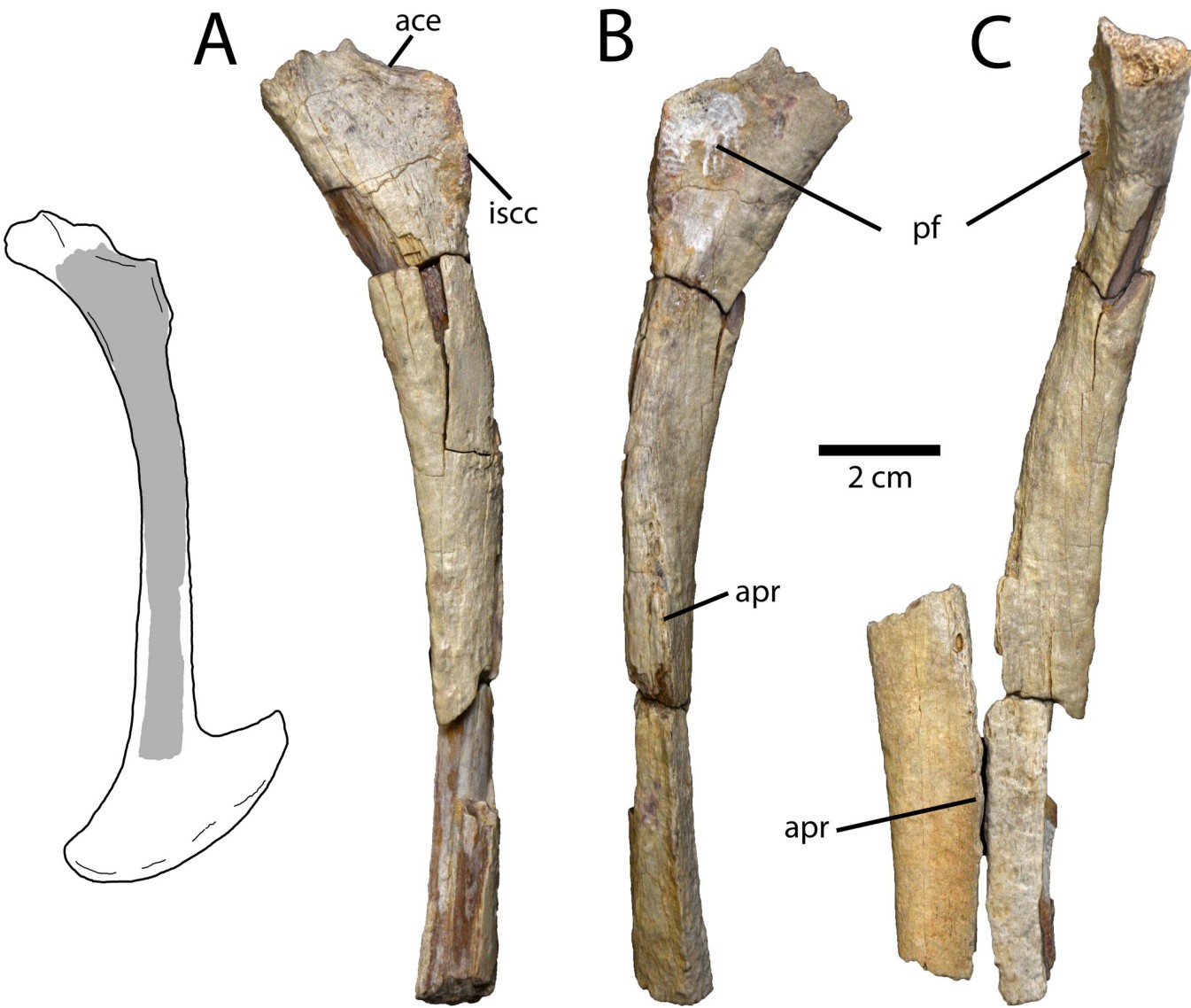

**Fig 4. Partial pubes of *Elmisaurus rarus* (MPC-D 102/113). A, B,** Left pubis in lateral (A) and medial (B) views. **C,** Left and right pubes in anterior view. Interpretive illustration of the pubes of "*Nomingia gobiensis*" (MPC-D 100/119) shows the preserved portions in MPC-D 102/113. **Abbreviations: ace,** acetabulum; **apr,** pubic apron; **iscc,** ischiadic contact; **pf,** pubic fossa.

**Astragalus.** A small portion of the medial condyle and ascending process of the left astragalus is preserved (Fig 5). Most of the ascending process is broken, but there is a shallow transverse sulcus at its base, and a depression dorsal to this. The articular surface of the medial condyle is rounded and has a finished, rather than a porous surface. The union of the ascending process and the condylar body forms an angle of 90° where the tibia would have inserted. Unfortunately, none of the contact with the calcaneum is preserved.

**Metatarsus.** All three of the weight-bearing metatarsals are preserved, but from differing sides. The right Metatarsal II is represented by the proximal end and a portion of the midshaft (Fig 6A–6E, 6H, 6I). The proximal end of the left Metatarsal III is preserved (Fig 6F and 6G), as well as the distal end of the right Metatarsal IV (Fig 6J–6N). The distal tarsals are missing; the clean proximal surfaces of the metatarsals suggest that no coossification of the distal tarsals

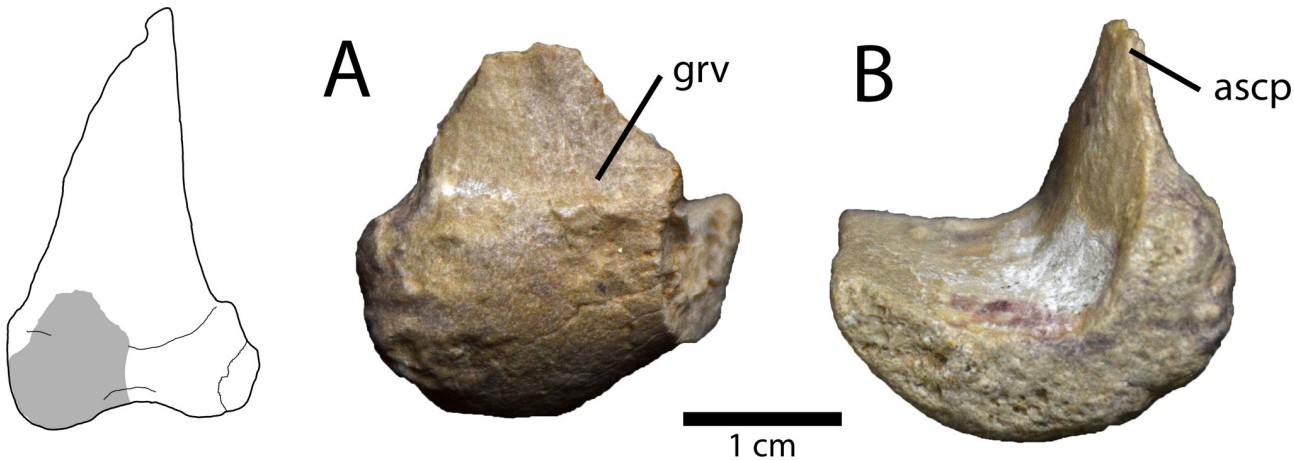

**Fig 5. Partial astragalus of *Elmisaurus rarus* (MPC-D 102/113). A, B,** Partial left astragalus in anterior (A) and medial (B) views. Interpretive illustration of "*Nomingia gobiensis*" (MPC-D 100/119) shows the preserved portion in MPC-D 102/113. **Abbreviations: ascp**, ascending process; **grv**, transverse groove.

or metatarsals had begun, but this is likely the result of ontogeny, as variation in fusion is known in *Elmisaurus rarus* [35]. The proximal end of the right Metatarsal II is somewhat semi-circular in proximal view (Fig 6E). On the lateral surface, there is a small anterior facet for Metatarsal IV and a recessed posterior facet for Metatarsal III. The posterior side has a square, protruding buttress with a ventrolaterally-inclined ventral edge. This buttress forms a part of the distinctive posterior protuberance that characterizes the tarsometatarsus of *Elmisaurus rarus* [35, 51]. The shaft of Metatarsal II has a well-developed posteromedial ridge and a small part of the facet for Metatarsal III. Separating these is a anterolaterally–posteromedially inclined posterolateral surface that would have contributed to the deep concavity in the plantar surface of the metatarsus. The proximal end of Metatarsal III is small and trapezoidal in cross-section (Fig 6F and 6G). The medial face curves anterolaterally where it would have contacted Metatarsal II, but it is broken. The anterior surface is triangular and has a small, ventrally-tapering flat area where it would have been exposed anteriorly. The posterior surface is antero-medially inclined and has a flat face that contributed to the posterior protuberance (Fig 6F). The right Metatarsal IV preserves the distal end and part of the shaft (Fig 6J–6N). The medial surface of the shaft has two ridges that outline the facet for Metatarsal III (Fig 6M). Unlike in ornithomimids and oviraptorids, this facet does not cover the entire medial face, and this would have contributed to the plantar concavity of the tarsometatarsus. The distal condyle of Metatarsal IV is small and bulbous, facing directly distally, rather than being deflected laterally. Both of the ligament pits are well-developed, and the shallower lateral one is bordered posteriorly by a wing-like triangular process (Fig 6N).

## Discussion

### Identity

MPC-D 102/113 can be identified as an oviraptorosaur on the basis of the laterally-everted acromion process of the scapula and by the pubis, which is mesopubic and anteriorly curved. This identity is further supported by the distinctive morphologies of the angular and the meta-tarsals. The skeleton contrasts with those of avimimids by being significantly larger than would be expected of *Avimimus nemegtensis* Funston et al. 2018 [14, 52–54], as well as the retention of the proximal end of Metatarsal III, which is lost in avimimids. Furthermore, the

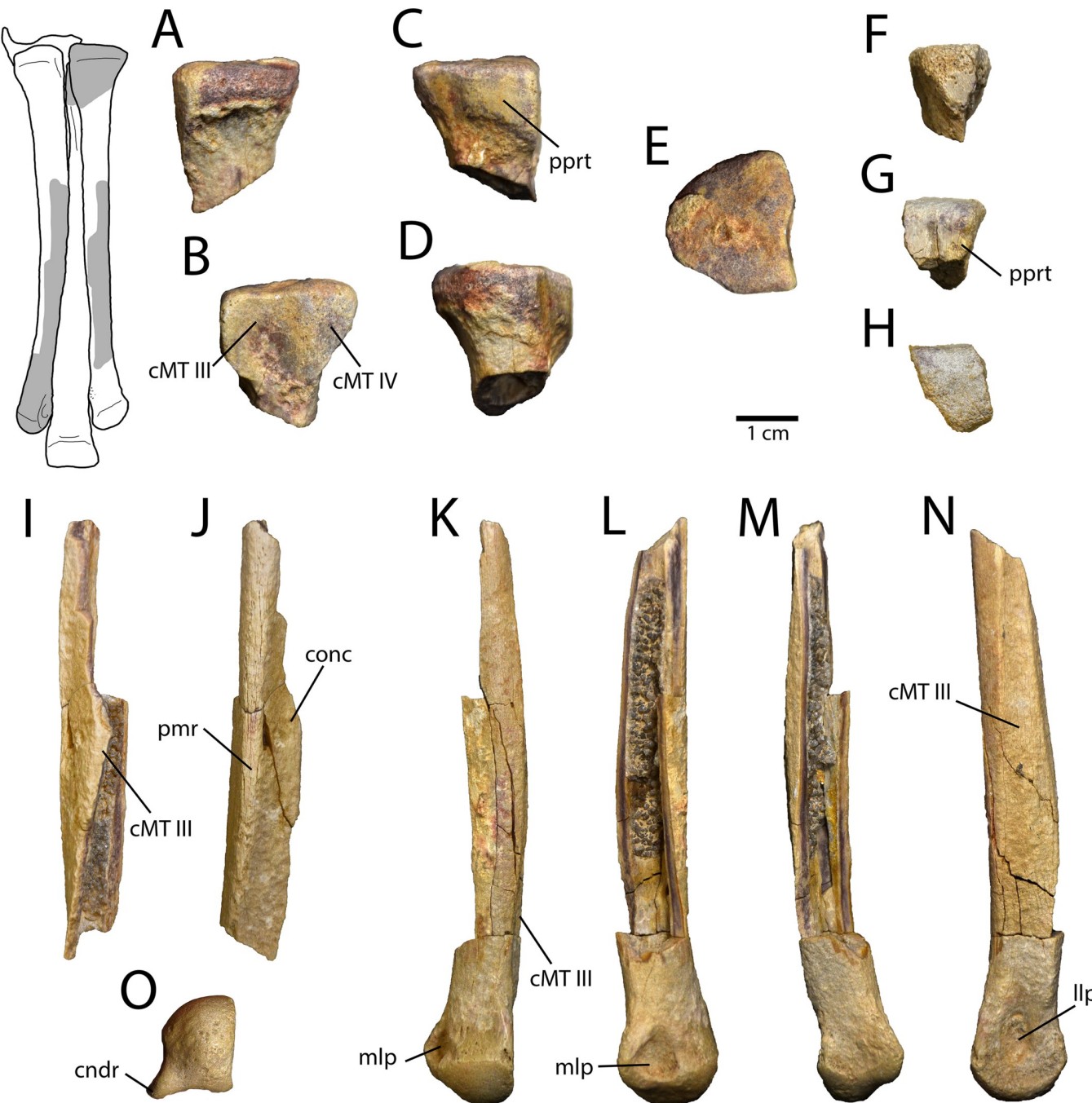

**Fig 6. Pedal elements of *Elmisaurus rarus* (MPC-D 102/113). A–E**, Proximal end of right metatarsal II in anterior (A), lateral (B), posterior (C), medial (D), and proximal (E) views. **F–H**, Proximal end of left metatarsal III in anterior (F), posterior (G), and proximal (H) views. **I, J**, partial shaft of right metatarsal II in lateral (I) and posterior (J) views. **K–O**, distal end of right metatarsal IV in anterior (K), lateral (L), posterior (M), medial (N), and distal (O) views. Interpretive illustration of tarsometatarsus of *Elmisaurus rarus* (MPC-D 102/006) shows the preserved regions in MPC-D 102/113. **Abbreviations: cMT III**, contact surface for metatarsal III; **cMT IV**, contact surface for metatarsal IV; **cndr**, condylar ridge; **llp**, lateral ligament pit; **mlp**, medial ligament pit; **pprt**, posterior protuberance.

preserved fragments of Metatarsals II and IV of MPC-D 102/113 indicate a deep plantar con-cavity on the tarsometatarsus, which is absent in both avimimids and oviraptorids. Compared to oviraptorids [26, 55, 56], the scapula is more robust and extends anteriorly to the acromion

process, which changes the relative positions of the glenoid and acromion process. Whereas these are more closely placed in oviraptorids, in MPC-D 102/113, the glenoid is anterior to the acromion process. The coracoid similarly differs from those of oviraptorids in the more dorsal positions of the biceps tubercle and coracoid foramen relative to the glenoid. The pubis also contrasts with those of oviraptorids in being relatively straight distally instead of anteriorly concave, and in the presence of a distinct, enclosed medial fossa at the proximal end [27], a feature that is less well developed in oviraptorids.

Instead, these features are more similar to caenagnathids. This interpretation is supported by the preserved proximal end of Metatarsal III, which is anteriorly pinched, indicating the unique semi-arctometatarsalian condition synapomorphic of caenagnathids. Among caenagnathids, several lines of evidence indicate that MPC-D 102/113 pertains to *Elmisaurus rarus*. Although the scapula is poorly known among caenagnathids, that of MPC-D 102/113 is similar to *Elmisaurus rarus* (ZPAL MgD-I/98) in the laterally-directed glenoid, which is positioned far anterior to the acromion process [35]. The proximal end of the pubis is consistent with ZPAL MgD-I/98 in the medially lipped contact for the ischium (Fig 7A and 7B), but this is also shared with *Nomingia gobiensis* (MPC-D 100/119). However, the strongest support for the identification of MPC-D 102/113 as *Elmisaurus rarus* is the distinctive combination of a deeply concave plantar surface of the metatarsus and the prominent tripartite posterior protuberance of the metatarsals at their proximal ends, which are shared with all known specimens of *Elmisaurus rarus* [22, 35]. In the closely-related *Citipes elegans*, the posterior protuberance and plantar concavity are much less well-developed, and in most other caenagnathids, the posterior protuberance is absent. Thus, the combination of these features is unique to *Elmisaurus rarus* within Caenagnathidae, and thus their presence in MPC-D 102/113 strongly argues in favour of its referral to this taxon.

## Taxonomic status of *Nomingia gobiensis*

Two caenagnathid taxa have historically been identified in the Nemegt Formation: *Elmisaurus rarus* and *Nomingia gobiensis*. However, little overlap exists between parts of the skeletons referred to these taxa, and elements that do overlap between the taxa are virtually identical [14], although none of these bones are considered highly diagnostic for caenagnathids. *Elmisaurus rarus* is represented by more numerous specimens, but these tend to be less complete than the holotype specimen of *Nomingia gobiensis* (MPC-D 100/119). However, since its initial description, the holotype of *Elmisaurus rarus* (ZPAL MgD-I/98), which remains the most complete specimen of this species, has undergone further preparation (Fig 7). This has revealed parts of the sacrum and scapula, as well as the ilium, pubis, ischium, femur, tibia, and fibula, and shows that these latter bones are nearly identical to those of *Nomingia gobiensis* [14]. Significantly, ZPAL MgD-I/98 and MPC-D 100/119 share an enlarged antitrochanter on the ischiadic peduncle, a medially protruding symphysis of the pubis and ischium, and a well-developed accessory trochanteric ridge on the femur. Beyond the type specimen of *Elmisaurus rarus*, most other known specimens are fragmentary. The most complete of these is a partial skeleton (MPC-D 102/007) with a frontal, some vertebrae, a hand, and a partial hindlimb. Vertebrae of this specimen compare well with the anterior dorsal vertebrae of *Nomingia gobiensis*, and the complete tibiae are virtually identical [14]. MPC-D 102/113 reveals further similarities in the skeletons of *Elmisaurus rarus* and *Nomingia gobiensis*, particularly in the curvature of the pubes, which is known to vary in caenagnathids [57]. It thus adds to a growing body of evidence that specimens from these two taxa represent a single morphotype. Furthermore, many of the features initially used to diagnose *Nomingia gobiensis* [24] can now be shown to be more widely distributed in Caenagnathidae. The short tail with large chevrons and a dorsally convex,

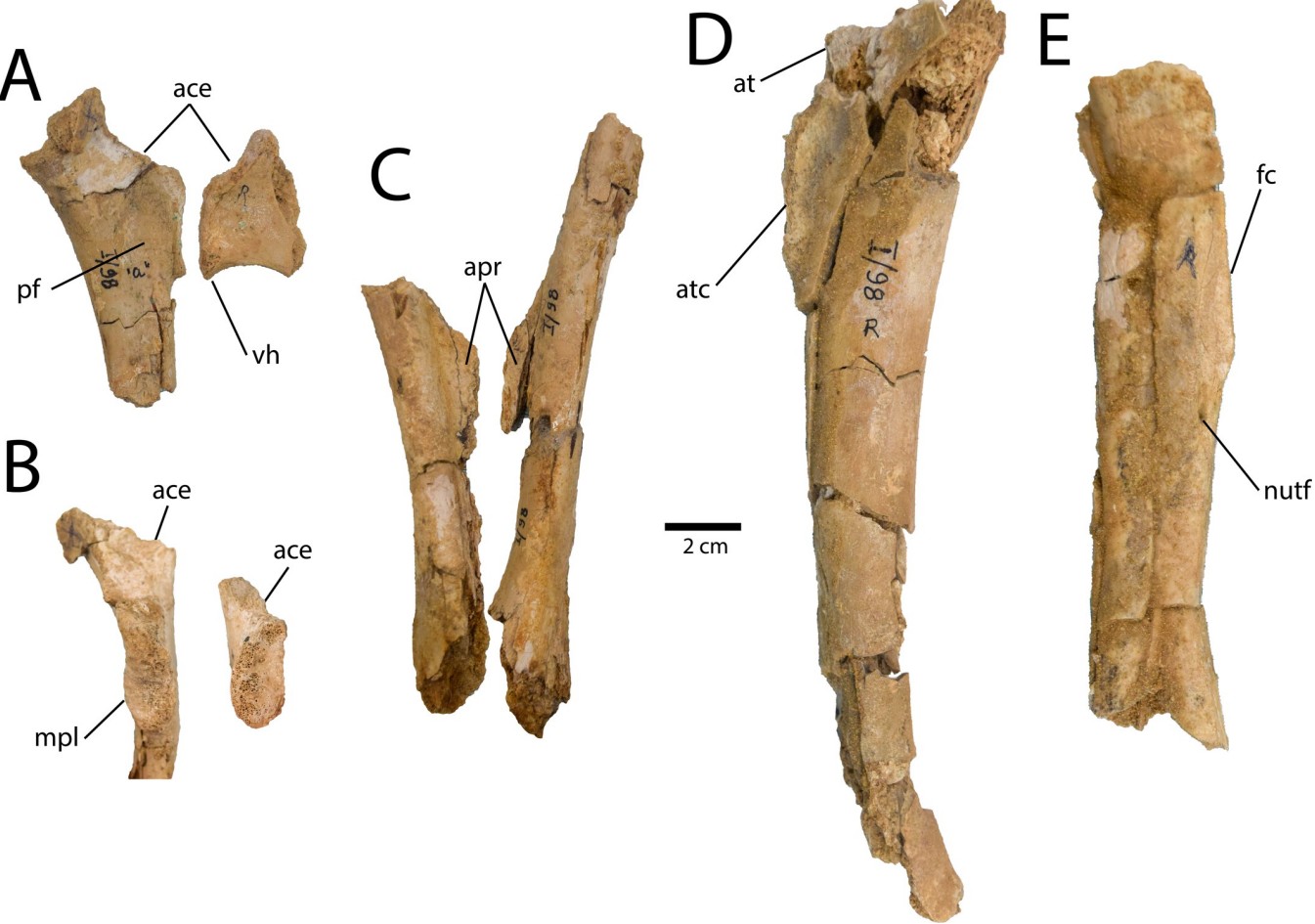

**Fig 7. Additional elements of *Elmisaurus rarus* (ZPAL MgD I/98) after further preparation. A, B,** Right proximal pubis (left) and ischium (right) in medial (A) and posterior (B) views. **C,** left and right pubic shafts in posterior view. **D,** right femur in medial view. **E,** right tibia shaft in posterior view. **Abbreviations: ace,** acetabulum; **apr,** pubic apron; **at,** anterior trochanter; **atc,** accessory trochanteric crest; **fc,** fibular crest; **mpl,** medially protruding lip; **nutf,** nutrient foramen; **pf,** pubic fossa; **vh,** ventral hook.

dolichoiliac ilium are present in *Chirostenotes pergracilis* [46, 48]; the weakly propubic pelvis is characteristic of most caenagnathids [27, 28, 57]; and the pubes are identical to those of *Elmisaurus rarus* [14].

It is particularly noteworthy that the discovery of an *Elmisaurus rarus* skeleton adjacent to the only known specimen of *Nomingia gobiensis* indicates that these taxa overlapped temporally and geographically. This is also supported by the recovery of an isolated pygostyle at Nemegt, which Persons et al. [34] referred to *Nomingia gobiensis*. This eliminates species turnover and geographic or temporal separation as valid explanations for the delineation of two taxa.

While none of these lines of evidence definitively show that *Nomingia gobiensis* is the same taxon as *Elmisaurus rarus*, they necessitate an unusual scenario where two taxa with virtually identical proportions, body size, and morphologies coexisted. Whereas caenagnathids are known to coexist elsewhere (e.g. the Dinosaur Park and Horseshoe Canyon Formations), taxa in these ecosystems are highly disparate in morphology and body size [29, 33]. Indeed, several studies have suggested that high oviraptorosaur diversity in several ecosystems was the result of morphological niche segregation [14, 15, 33, 58]. Considering the low likelihood of this

coexistence scenario and the lack of positive evidence for two morphotypes, "*Nomingia gobiensis*" is best regarded as junior subjective synonym of *Elmisaurus rarus*. Confirmation or refutation of the synonymy of these taxa could come from a discovery of an ilium referable to *Elmisaurus rarus* or a tarsometatarsus referable to "*Nomingia gobiensis*", as these elements are highly diagnostic in oviraptorosaurs.

## Anatomy of *Elmisaurus rarus*

Assuming that material of "*Nomingia gobiensis*" is actually representative of *Elmisaurus rarus*, it would become one of the most completely known caenagnathids (Fig 8). In any case, MPC-D 102/113 provides more information on the anatomy of *Elmisaurus rarus*, particularly the scapulocoracoid and pubes, which were poorly known previously.

The possible postorbital provides some tentative clues to the cranial morphology of *Elmisaurus rarus*. If the interpretation presented here is correct, the squamosal process would have been short, indicating a relative small infraorbital fenestra. The lateral placement of the facet for the squamosal is also unusual, and may indicate an interfingering contact with the squamosal. However, the crania of caenagnathids are exceptionally poorly known, and therefore this element may have been misidentified. It also bears some semblance to the posterior portion of the maxilla, and it could conceivably represent part of the palate as well. As future discoveries elucidate more of the caenagnathid skull, reinterpretation of this element may provide more information on the cranial anatomy of *Elmisaurus rarus*.

The scapulocoracoid of *Elmisaurus rarus* exhibits several unusual features compared to other caenagnathids and oviraptorosaurs [14, 15, 26]. In particular, the position of the glenoid anterior to the acromion process is unusual compared to most other oviraptorosaurs, especially oviraptorids, although this feature appears to be similar in *Apatoraptor pennatus* as well [29]. This may indicate a stronger relationship between *Apatoraptor pennatus* and *Elmisaurus rarus*, or it could be more widely distributed in Caenagnathidae, scapulae of which are poorly known. The posteroventral process of the coracoid is less tightly curved than that of *Chirostenotes pergracilis* [48], and in this respect it also compares more closely to *Apatoraptor pennatus* [29]. The unusual morphologies of the scapulae of *Apatoraptor pennatus* and *Elmisaurus rarus* suggest a divergent function from other oviraptorosaurs. As discussed by Funston and Currie [29], a more laterally oriented glenoid would have enabled greater anterior extension of the forelimbs. The coracoid also exhibits some unusual features, notably the more dorsally positioned biceps tubercle and coracoid foramen. These features may be associated with a rearrangement of the musculature and vasculature necessitated by the unusual position and orientation of the scapular portion of the glenoid. Together, these modifications produce an expanded anterodorsal region of the scapulocoracoid, which probably accommodated large origins for the forelimb protractors *m. supracoracoideus* and *m. deltoideus* [59, 60]. The development of these muscles combined with the more laterally oriented glenoid would have enabled more powerful and greater protraction of the forelimb than other oviraptorosaurs were capable of. Considering the elongate arms of *Apatoraptor pennatus* and the raptorial manus of both *Apatoraptor pennatus* and *Elmisaurus rarus*, this is possibly indicative of a prominent prey capture role for the forelimb, although other functions are plausible as well.

The morphology of the pubis of *Elmisaurus rarus* is generally consistent with other caenagnathids in that it is distally straight, rather than anteriorly concave, and has a transversely narrow pubic apron [26–28, 57, 61]. The proximal end bears an enclosed medial fossa, described by Sullivan et al. [27] as a synapomorphy of caenagnathids. The only distinctive feature of the pubis of *Elmisaurus rarus* is that the contact for the ischium protrudes medially, a feature that it shares with "*Nomingia gobiensis*" (MPC-D 100/119) and some small indeterminate pubes

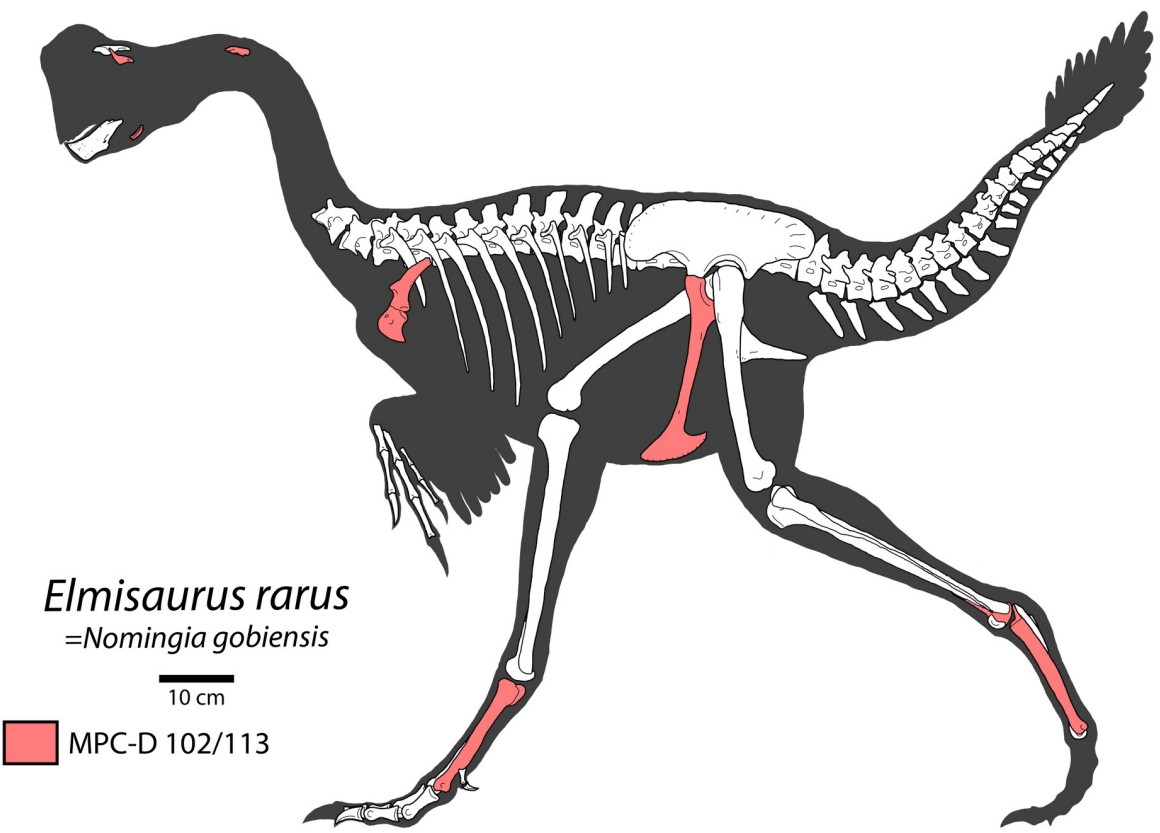

**Fig 8. Skeletal reconstruction of *Elmisaurus rarus*.** Skeletal reconstruction of known elements of *Elmisaurus rarus*, including material referred to *"Nomingia gobiensis"*. Elements highlighted in red are preserved in MPC-D 102/113.

from the Dinosaur Park Formation of Alberta, Canada [57]. The pubis of MPC-D 100/119 (*"Nomingia gobiensis"*) is unusual among oviraptorosaurs in that the posterior process of the pubic boot is equal in length to the anterior process. Most oviraptorosaurs are distinguished by a posterior process of the pubic boot that is distinctly shorter than the anterior process [26, 57]. If this distinctive morphology were discovered in an indisputable *Elmisaurus rarus*, it would provide further support of the synonymy of *Elmisaurus rarus* and *"Nomingia gobiensis"*.

The metatarsals of MPC-D 102/113 do not show evidence of proximal fusion, a feature which typically distinguishes *Elmisaurus rarus* from most other caenagnathids. However, this is likely attributable to ontogeny rather than taxonomy, as fusion of compound elements tends to increase with age in most dinosaurs [53, 62]. For example, known tarsometatarsi of *Elmisaurus rarus* exhibit considerable variation in the extent of proximal coossification of the distal tarsals and metatarsals. As described by Currie et al. [35], fusion appears to proceed from posterior to anterior, and proximal to distal. The earliest sign of coossification is the adhesion of distal tarsal III to the proximal surface of Metatarsal II, as exhibited in MPC-D 102/007 [35]. MPC-D 102/113 had apparently not yet reached this stage of maturity. Eventually, distal tarsals III and IV coossify with each other and with the posterior protuberance of the metatarsals (as in MPC-D 102/006), but the order in which these events occurred is unknown. Finally, the distal tarsals and the proximal ends of the metatarsals become indistinguishable, although a suture persists anteriorly between Metatarsals II and IV (as in ZPAL MgD-I/98 and ZPAL MgD-I/172; Osmólska, 1981). However, other features of the metatarsals supporting the identification of the specimen as *Elmisaurus rarus*, like the deeply concave plantar surface of the

metatarsus and the prominent tripartite posterior protuberance on the proximal end of the metatarsus, are unlikely to have been significantly affected by ontogeny. The deeply concave plantar surface of the metatarsus is formed by the cruciate ridges on the anteroposteriorly flattened Metatarsal III and the prominent posteromedial and posterolateral ridges on Metatarsals II and IV, respectively. Each of these features are present in all of the metatarsi attributable to *Elmisaurus rarus*, regardless of size and extent of fusion, suggesting that they do not vary systematically through ontogeny. Importantly, the posteromedial and posterolateral ridges of Metatarsals II and IV are more robust and extend further proximally along the shaft of the metatarsals than in other adult caenagnathids [33], suggesting that their larger size is unique to *Elmisaurus rarus*. Similarly, the presence of a posterior protuberance formed by the proximal metatarsals in MPC-D 102/113 and MPC-D 102/006 shows that this distinctive feature is present prior to coossification of the tarsometatarsus. A similar, but less-well developed feature has been described in metatarsals attributed to *Caenagnathus collinsi* from the Dinosaur Park Formation [63], but it is unclear whether the metatarsals fused in this animal. Conversely, in the small-bodied *Citipes elegans* (Parks 1933) [33, 51, 64], the metatarsals are proximally fused and presumably attributable to adults [33], but the tripartite protuberance is much less prominent than in *Elmisaurus rarus* [51, 65]. Thus, it appears that this more prominent feature in *Elmisaurus rarus* is also better explained as a taxonomic character rather than an ontogenetic one. The robustly fused metatarsals, plantar concavity, and posterior protuberance may be evidence of powerful digit adductor musculature in *Elmisaurus rarus*. An in depth analysis of pedal musculature is beyond the scope of this study, but future study of the tarsometatarsus of *Elmisaurus rarus* may prove fruitful for understanding its palaeoecology. Likewise, systematic osteohistological sampling [33, 66] of *Elmisaurus rarus* specimens could illuminate the ontogenetic trends of fusion and other characters in more detail.

## Relationships of *Elmisaurus rarus*

The synonymy of Caenagnathidae and Elmisauridae is now fairly unambiguous, and there is little evidence to suggest that 'elmisaurines' form a monophyletic clade within Caenagnathidae. Regardless, there has previously been little consensus on the relationships of *Elmisaurus rarus* within Caenagnathidae. Recent analyses ally it with either *Apatoraptor pennatus* or *Citipes elegans* from North America [15, 28, 29]. MPC-D 102/113 highlights some characters that seem to strengthen the relationship of *Apatoraptor pennatus* to *Elmisaurus rarus*. For example, the angulars of each taxon bow ventrally, as is also the case in *Chirostenotes pergracilis*, but they also have a more deeply incised channel for the posteroventral ramus of the dentary. In *Anzu wyliei*, *Caenagnathus collinsi*, and *Chirostenotes pergracilis*, in contrast, the groove for the dentary becomes shallower posteriorly, without a dorsally overhanging lip. In any case, the similarity between the angulars of *Apatoraptor pennatus*, *Chirostenotes pergracilis*, and *Elmisaurus rarus* lends support to the idea that the mandibles would have been similar, and thus that the dentaries described by Tsuihiji et al. [36] are referable to the latter taxon. This interpretation would be further strengthened if more evidence of the synonymy of *Elmisaurus rarus* and "*Nomingia gobiensis*" were uncovered. Features of the scapulocoracoid are also shared between *Apatoraptor pennatus* and *Elmisaurus rarus*, particularly the more laterally oriented glenoid and the straighter posteroventral process of the coracoid. These taxa are also distinctive among oviraptorosaurs in sharing the anteriorly elongated scapula, providing increased room for supracoracoid and deltoid musculature. Funston and Currie [46] described a partial tail of *Chirostenotes pergracilis*, noting several similarities to "*Nomingia gobiensis*". In particular, the specimens share the distinctive anteroposteriorly elongate chevrons, and the anteriorly directed transverse processes of the distal caudal vertebrae. However,

the conditions of these characters are completely unknown in *Citipes elegans*, and it is possible that they were shared more widely within caenagnathids. Furthermore, chevron morphology can be highly disparate within the families of Oviraptorosauria (e.g. *Heyuannia yanshini* and the Zamyn Khondt oviraptorid) or even a single taxon [26, 67], and thus shared chevron shape may not be a strong indicator of a close relationships. Nevertheless, several features unite *Elmisaurus rarus* with other small-bodied, Campanian–Maastrichtian caenagnathids from North America. This suggests that its distribution in Mongolia is likely the result of dispersal from North America in the Campanian–Maastrichtian, rather than representing an offshoot of an Asian lineage [15, 68].

## Caenagnathid diversity in the Nemegt Formation and Asia

The synonymy of *"Nomingia gobiensis"* and *Elmisaurus rarus* suggests that only a single caenagnathid taxon was present in the Nemegt Formation. This is a lower species richness than many other caenagnathid-producing geological formations, especially those of the Campanian–Maastrichtian of western North America, where two or three genera typically coexist [28, 29, 33, 69, 70]. As caenagnathids are rare elements of the faunae to which they belong, it could be argued that this is simply an artifact of sampling, and that low species richness in the Nemegt Formation is attributable to poor sampling. However, this does not seem to be the case, as several localities within the Nemegt Basin are excellently sampled [14, 20] and have produced numerous small theropod skeletons. Furthermore, *Elmisaurus rarus* is known from more skeletons than other caenagnathids, and these are typically of higher completeness than in North America, where the caenagnathid record is composed predominantly of isolated bones [27, 33, 37, 70, 71]. Thus, it appears that the low diversity of caenagnathids in the Nemegt Formation is not the result of poor sampling. Instead, this discrepancy may be attributable to a notable difference between Asian and North American ecosystems: the presence of other oviraptorosaur families. Whereas caenagnathids were the only oviraptorosaurs in North America, both avimimids and oviraptorids coexisted with caenagnathids in the Nemegt Formation. Oviraptorids in particular were highly diverse in the Nemegt Formation [14, 15, 72] and although they were specialized for different diets [58, 73], they may still have competed with caenagnathids for niche space as similarly-sized omnivores. The same may be true of other Late Cretaceous ecosystems in Asia, as several formations like the Bayan Mandahu, Djadokhta, and Nanxiong Formations have high diversities of oviraptorids and other small theropods, but have not yet produced any caenagnathids, despite the clear presence of caenagnathids in Asia by this time [37, 74–78]. Indeed, the caenagnathid record precedes that of oviraptorids in Asia [74, 78], but there are no examples of coexistence between earlier endemic Asian caenagnathids and oviraptorids. Although tentative, this coexistence pattern suggests that oviraptorids may have displaced endemic Asian caenagnathids during the Campanian. North American caenagnathids were able to redistribute to Asia later in the Cretaceous [33], but it appears that they remained marginalized and were not able to diversify as effectively as in North America.

## Conclusions

A new, partial skeleton from the Nemegt Formation of Bügiin Tsav provides some additional information on the skull, mandible, and pectoral girdle of *Elmisaurus rarus*. This specimen further highlights similarities in the overlapping elements of *Elmisaurus rarus* and *"Nomingia gobiensis"*, supporting the idea that the latter is the junior synonym of the former. The unusual pectoral girdle of *Elmisaurus rarus* suggests a close affinity with *Apatoraptor pennatus* from the Horseshoe Canyon Formation of Alberta, Canada, as well as an increased capability for

powerful extension of the forelimbs. Low diversity of caenagnathids in the Nemegt Formation may reflect broader marginalization of caenagnathids in Asia during the Late Cretaceous, possibly as a result of competition with oviraptorids or other oviraptorosaurs.

## Acknowledgments

We thank Sanjaadash Ulziitseren for access to the collections at the Institute of Palaeontology of the Mongolian Academy of Sciences. Fieldwork was made possible by National Geographic funding to Federico Fanti, who is thanked for logistical support. We thank M. Pittman and T. Holtz, Jr., who provided helpful, constructive reviews that improved the manuscript.

## Author Contributions

**Conceptualization:** Gregory F. Funston, Philip J. Currie, Chinzorig Tsogtbaatar, Tsogtbaatar Khishigjav.

**Data curation:** Philip J. Currie, Chinzorig Tsogtbaatar, Tsogtbaatar Khishigjav.

**Formal analysis:** Gregory F. Funston.

**Investigation:** Gregory F. Funston, Philip J. Currie, Chinzorig Tsogtbaatar.

**Methodology:** Gregory F. Funston.

**Project administration:** Gregory F. Funston.

**Resources:** Tsogtbaatar Khishigjav.

**Visualization:** Gregory F. Funston.

**Writing – original draft:** Gregory F. Funston.

**Writing – review & editing:** Gregory F. Funston, Philip J. Currie, Chinzorig Tsogtbaatar, Tsogtbaatar Khishigjav.

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
