## [Decision Letter · Decision Letter 0]

10 Jun 2021

PONE-D-21-13378

A partial caenagnathid skeleton from the Nemegt Formation of Bügiin Tsav, and the status of *Nomingia gobiensis* (Theropoda: Oviraptorosauria)

PLOS ONE

Dear Dr. Funston,

Thank you for submitting your manuscript to PLOS ONE. After careful consideration, we feel that it has merit but does not fully meet PLOS ONE’s publication criteria as it currently stands. Therefore, we invite you to submit a revised version of the manuscript that addresses the points raised during the review process.

Please address the reviewers issues, especially the diversity question proposed by Dr. Pittman. Beyond that there are only very minor issues to be dealt with before this manuscript can be accepted for publications. This is a quality work and I believe that it will be a strong addition to the field. 

We look forward to receiving your revised manuscript.

Kind regards,

T. Alexander Dececchi, Ph.D

Academic Editor

PLOS ONE

Journal Requirements:

6. We note that Figure 1 in your submission contain satellite images which may be copyrighted. All PLOS content is published under the Creative Commons Attribution License (CC BY 4.0), which means that the manuscript, images, and Supporting Information files will be freely available online, and any third party is permitted to access, download, copy, distribute, and use these materials in any way, even commercially, with proper attribution. For these reasons, we cannot publish previously copyrighted maps or satellite images created using proprietary data, such as Google software (Google Maps, Street View, and Earth). For more information, see our copyright guidelines: http://journals.plos.org/plosone/s/licenses-and-copyright.

7. We note that Figure 2, 3, 4, 5, 6, 7 and 8 in your submission contain copyrighted images. All PLOS content is published under the Creative Commons Attribution License (CC BY 4.0), which means that the manuscript, images, and Supporting Information files will be freely available online, and any third party is permitted to access, download, copy, distribute, and use these materials in any way, even commercially, with proper attribution. For more information, see our copyright guidelines: http://journals.plos.org/plosone/s/licenses-and-copyright.

a. You may seek permission from the original copyright holder of Figure 2, 3, 4, 5, 6, 7 and 8 to publish the content specifically under the CC BY 4.0 license.

Additional Editor Comments (if provided):

First off congratulations on a high quality manuscript. Both reviewers enjoyed and recommended this work with only minor issues. I agree with both of them and suggest you follow their recommendations especially the issue about the use of nomen dubium raised by Dr. Holtz (which can be easily dealt with with minor text alterations) and the issue raised by Dr. Pittman about low diversity of caenagnathids in the Nemegt compared to other oviraptorosaurs and how this compares to other well studied regions. I do agree such a broader picture perspective would increase the readership and impact of this paper and, given your strong background and work in this area, I do not believe that compiling this data would be much of a stretch. I feel this small addition, a small analysis and a couple of paragraphs in the discussion, would greatly increase this manuscripts influence on future discussion both on oviraptorosaur diversity trends and other clades. I look forward to reading the next version as I think with those small modifications and the other very minor comments the two reviewers bring up this paper would be more than acceptable for publication in PLoS one.

Hope all is well and once again congratulations to all the authors on your hard work.

TA Dececchi

Reviewers' comments:

Reviewer's Responses to Questions

**Comments to the Author**

1. Is the manuscript technically sound, and do the data support the conclusions?

Reviewer #1: Yes

Reviewer #2: Yes

2. Has the statistical analysis been performed appropriately and rigorously? 

Reviewer #1: Yes

Reviewer #2: N/A

3. Have the authors made all data underlying the findings in their manuscript fully available?

Reviewer #1: Yes

Reviewer #2: Yes

4. Is the manuscript presented in an intelligible fashion and written in standard English?

Reviewer #1: Yes

Reviewer #2: Yes

5. Review Comments to the Author

Reviewer #1: Dear Dr. Funston and colleagues (Hi Greg, Phil, Chinzorig and Tsogtbaatar!),

Thank you for the opportunity to review your work. I agree with your synonymy and think it is convincingly argued.

I have recommended a few references for you to add -- I think they would strengthen your paper. My main suggestion is to discuss what low caenagnathid species count in the Nemegt tells us about the local oviraptorosaur fauna e.g. were caenagnathids marginal members there? This naturally springboards onto whether the make up of oviraptorosaur faunas are similar in this way regionally and globally or if this was unique to the Nemegt. These are bigger picture questions that the reader would be very interested in that your work provides an ideal opportunity to comment on. I hope you can comment on these aspects.

Thanks for taking my comments on board. I look forward to seeing the manuscript published at the journal.

Best regards,

Dr. Michael Pittman

The University of Hong Kong

Reviewer #2: In a world where partial skull caps or proximal femora become the source of new taxon names, it is pleasant and encouraging to see the case where two relatively well-known sympatric species based on non-overlapping parts of the anatomy turn out to be just one taxon.

The anatomical descriptions of the specimens (both the new one, and the reprepared Elmisaurus rarus holotype) are thorough. The case uniting this specimen with both Nomingia gobiensis and Elmisaurus rarus is strong; those features where it differs from some specimens of Elmisaurus rarus (e.g., lack of fusion of distal tarsals and proximal ends of the metatarsals) are shown to be variable within Elmisaurus already.

I disagree with the authors, however, on one specific item on p. 19, line 406. I do not think that they have shown that Nomingia gobiensis is a nomen dubium. A nomen dubium isn’t simply an invalid name; it is specifically a reference to a species name in which the type specimen is in fact insufficient to diagnose the species in question (see Mones 1989, for instance). That isn’t the case here; if the name “Elmisaurus rarus” had not been proposed, it seems as if the type specimen of Nomingia gobiensis could have served as a valid source for an oviraptorosaur name distinct from other known named taxa. Instead, the authors have simply shown that Nomingia gobiensis is a junior subjective synonym of Elmisaurus rarus (“junior” in that it was named later; “subjective synonym” in that they refer to the same taxon but use different type specimens.)

Mones, A. 1989. Nomen dubium vs. nomen vanum. Journal of Vertebrate Paleontology 9: 232-234.

COMMENTS

Line 514 It might be appropriate to cite Persons et al. (2015) here, with the demonstration that chevron morphology can vary intraspecifically (sexually?) within an oviraptorosaur.

Persons, W.S., IV, G.F. Funston, P.J. Currie & M.A. Norell. 2015. A possible instance of sexual dimorphism in the tails of two oviraptorosaur dinosaurs. Scientific Reports 5: 9472. Doi: 10.1038/srep09472

6. PLOS authors have the option to publish the peer review history of their article (what does this mean?). If published, this will include your full peer review and any attached files.

Reviewer #1: **Yes: **Dr. Michael Pittman

Reviewer #2: **Yes: **Thomas R. Holtz, Jr.

---

## [Author Response · Author response to Decision Letter 0]

24 Jun 2021

Reviewer comments: 

Reviewer #1: Dear Dr. Funston and colleagues (Hi Greg, Phil, Chinzorig and Tsogtbaatar!),

Thank you for the opportunity to review your work. I agree with your synonymy and think it is convincingly argued.

Reponse: Thanks for your positivity about our manuscript! 

I have recommended a few references for you to add -- I think they would strengthen your paper. My main suggestion is to discuss what low caenagnathid species count in the Nemegt tells us about the local oviraptorosaur fauna e.g. were caenagnathids marginal members there? This naturally springboards onto whether the make up of oviraptorosaur faunas are similar in this way regionally and globally or if this was unique to the Nemegt. These are bigger picture questions that the reader would be very interested in that your work provides an ideal opportunity to comment on. I hope you can comment on these aspects.

Reponse: Those are great references to add, and we have done so. We drafted the manuscript prior to the publication of the book but should have updated our references after it came out. Your suggestion for incorporating bigger-picture implications is a good one, and we have added a new discussion section building on the implication of low diversity in the Nemegt, expanding this out to some reasonable speculation based on the fossil record of caenagnathids in Asia. We feel that this definitely raises some preliminary but testable ideas that will be of interest to a broad readership. We have added some brief statements summarizing these ideas to the abstract and conclusions sections.

Thanks for taking my comments on board. I look forward to seeing the manuscript published at the journal.

Reponse: Thanks for your effort in annotating the manuscript. We have not responded individually to your comments on the marked-up manuscript here, but we have incorporated them all into the manuscript. 

Reviewer #2: In a world where partial skull caps or proximal femora become the source of new taxon names, it is pleasant and encouraging to see the case where two relatively well-known sympatric species based on non-overlapping parts of the anatomy turn out to be just one taxon.

Reponse: Thanks for your positivity about our manuscript!

The anatomical descriptions of the specimens (both the new one, and the reprepared Elmisaurus rarus holotype) are thorough. The case uniting this specimen with both Nomingia gobiensis and Elmisaurus rarus is strong; those features where it differs from some specimens of Elmisaurus rarus (e.g., lack of fusion of distal tarsals and proximal ends of the metatarsals) are shown to be variable within Elmisaurus already.

I disagree with the authors, however, on one specific item on p. 19, line 406. I do not think that they have shown that Nomingia gobiensis is a nomen dubium. A nomen dubium isn’t simply an invalid name; it is specifically a reference to a species name in which the type specimen is in fact insufficient to diagnose the species in question (see Mones 1989, for instance). That isn’t the case here; if the name “Elmisaurus rarus” had not been proposed, it seems as if the type specimen of Nomingia gobiensis could have served as a valid source for an oviraptorosaur name distinct from other known named taxa. Instead, the authors have simply shown that Nomingia gobiensis is a junior subjective synonym of Elmisaurus rarus (“junior” in that it was named later; “subjective synonym” in that they refer to the same taxon but use different type specimens.)

Mones, A. 1989. Nomen dubium vs. nomen vanum. Journal of Vertebrate Paleontology 9: 232-234.

Reponse: Yes, this is absolutely correct and we have changed it in the manuscript. Thanks for catching that! 

COMMENTS

Line 514 It might be appropriate to cite Persons et al. (2015) here, with the demonstration that chevron morphology can vary intraspecifically (sexually?) within an oviraptorosaur.

Persons, W.S., IV, G.F. Funston, P.J. Currie & M.A. Norell. 2015. A possible instance of sexual dimorphism in the tails of two oviraptorosaur dinosaurs. Scientific Reports 5: 9472. Doi: 10.1038/srep09472

Reponse: Yes, it is definitely an appropriate reference for that statement, and we have added it and refined the sentence to capture the sentiment that chevron morphology is variable both within and between taxa.

---

## [Editor Report · Decision Letter 1]

30 Jun 2021

A partial oviraptorosaur skeleton suggests low caenagnathid diversity in the Late Cretaceous Nemegt Formation of Mongolia

PONE-D-21-13378R1

Dear Dr. Funston,

We’re pleased to inform you that your manuscript has been judged scientifically suitable for publication and will be formally accepted for publication once it meets all outstanding technical requirements.

Kind regards,

T. Alexander Dececchi, Ph.D

Academic Editor

PLOS ONE

Additional Editor Comments:

I want to thank you all for listening to the reviewers comments and addressing them in your revision. I know adding a new part to the discussion increased the workload for you, but I hope you agree that it does improve the impact and reach of your paper. I look forward to seeing this "on the shelves" soon. Great job to you and your team.

---

## [Editor Report · Acceptance letter]

2 Jul 2021

PONE-D-21-13378R1 

A partial oviraptorosaur skeleton suggests low caenagnathid diversity in the Late Cretaceous Nemegt Formation of Mongolia 

Dear Dr. Funston:

I'm pleased to inform you that your manuscript has been deemed suitable for publication in PLOS ONE. Congratulations! Your manuscript is now with our production department. 

Kind regards, 

on behalf of

Dr. T. Alexander Dececchi 

Academic Editor

PLOS ONE